# The Assessment of Hydrogeosites in the Fann Mountains, Tajikistan as a Basis for Sustainable Tourism

**Katarzyna Pukowiec-Kurda** , **Oimahmad Rahmonov** , **Michał Sobala \*** and **Urszula Myga-Piątek**

Faculty of Natural Sciences, University of Silesia, 41200 Sosnowiec, Poland; katarzyna.pukowiec@us.edu.pl (K.P.-K.); oimahmad.rahmonov@us.edu.pl (O.R.); urszula.myga-piatek@us.edu.pl (U.M.-P.)
\* Correspondence: michal.sobala@us.edu.pl; Tel.: +48-32-3689-263

**Abstract:** Despite the fact that the Fann Mountains are among the most popular tourist destinations in Tajikistan, they are still in the first stage of tourism development. This represents a great opportunity for the implementation of the principles of sustainable tourism, which will avoid the mistakes associated with the uncontrolled tourism development currently observed in other mountain areas of the world. The aim of this article is to demonstrate, using the example of the Fann Mountains, how hydrogeosites in mountain areas can be valorised for the needs of cognitive tourism. The valorisation methods used in previous research to this point have focused on the evaluation of the objects themselves. This study additionally takes into account features of the surroundings of hydrogeostations, such as the visibility range, the vertical development of the view, and the diversity of the landscape. The conducted value assessments of the sites and their surroundings show that in both internal and external assessments the highest values were achieved by lakes and wetlands. This means that the evaluation of the surroundings has a strong influence on the results obtained and the choice of hydrogeotourism attractions.

**Keywords:** Tajikistan; high mountain tourism; geotourism; aesthetic value

## 1. Introduction

Today, global environmental changes are very dynamic, and one of the main factors causing them is human activity [1]. Such recent changes, including global warming, are being obfuscated [2,3]. Hence, education on global processes is vital because an educated society is less susceptible to disinformation campaigns [4]. While tourism is one human activity that can harm the environment, it may also be treated as a crucial factor in increasing social education about environmental problems. The educational function of tourism is widely used today and ecological awareness is shaped through communing with nature. This contact is one reason why we need to take care of the natural environment [5,6]. In the face of dynamic environmental changes caused by human activity, tourism should play an important educational role. It may have a positive influence because tourism combines entertainment with education, shaping attitudes and sensitivity towards contemporary environmental problems.

Today, one of the most crucial environmental problems that humanity needs to deal with is global warming [7]. One of the effects of climate change is changes in water resources [8,9]. These are particularly important in mountain environments characterised by high vulnerability to change [10]. The increasing global warming is contributing to the disappearance of mountain glaciers worldwide, which act as sources of water for mountain areas [11] and those surrounding them. For example, H. D. Pritchard [12] reports that glaciers are essential sources of fresh water in the western river basins of the high-mountain Asia region (Pakistan, Tajikistan, Uzbekistan, Kyrgyzstan, and Xinjiang in northwestern China), particularly during droughts. This is because these regions receive little precipitation during summer when water demand for irrigation is highest. Another

issue is that, as a result of melting mountain glaciers, Glacier Lake Outburst Floods (GLOFs) are becoming more common [13,14]. These high-energy events can be significant threats to human life and infrastructure, causing disasters along stretches of land further down valleys [15,16]. These issues prove that mountain areas characterised by a vulnerable environment are simultaneously of great economic and environmental importance. It must be emphasised that mountain areas occupy about 24% of the landmass surface worldwide and are inhabited by about 12% of the world's population. It is estimated that another 14% of the population lives in close vicinity to them [17].

Hence, mountain areas seem to be suitable for educational purposes concerning global environmental changes, especially those connected with water resources. Because the growing demand among tourists for new types of destinations and the increasing recognition of the importance of geological heritage have led to the spread of geotourism [18,19], it seems that geosites can be used for educational purposes in tourism. This paper aims to assess hydrogeosites in terms of their suitability for tourism purposes. For this reason, we chose one test area located in the Fann Mountains (Tajikistan) where the locations of hydrogeosites make the creation of a tourist route possible.

Although the most famous mountain destinations in Tajikistan are in the Fann Mountains [20], this area is still in the first stage of tourism development [21]. Such development is essential because the low degree of development of tourist infrastructure makes it possible to shape tourism in a sustainable way without conflicts connected with the various needs of tourists. This will help to avoid mistakes connected with the uncontrolled development of tourism that can be observed in other mountain areas such as the Indian Himalayas [7], the Alps [22], the Andes [23], and many others [24].

We chose to study hydrological geosites, which can be defined as a basic type of geotourist object connected with hydrology and hydrogeology that is a place of particular importance for learning about the history of the Earth, including the history of life and climate change [25]. These objects are part of a hydrological heritage. They stand out in terms of their environmental, scientific, educational, socio-cultural, and aesthetic values [26]. Hydrological heritage is a relatively new concept in the field of geosite assessment. There are many classifications of geosites and hydrogeologic geosites [27]. They generally include usual and karst springs, intermittent (rhythmic) springs, geothermal hot springs, and those with specific socio-cultural values, as well as rivers, lakes, ponds, and oxbow lakes [26]. Moreover, D. Mijović et al. [28] also consider mineral waters, geothermal hot water, submerged springs, and hydrogeological objects of historical significance to be geosites. L'. Štrba [29] also distinguishes anthropogenic geosites.

Multi-criteria valorisation of geosites is not a new issue. It has been the object of research for many authors in different parts of the world [30–40]. Different national geomorphological contexts and objectives have not allowed for the development of universal guidelines. Moreover, until now, studies have only focused on the features of the objects themselves, without taking into account the values of their surroundings. The features of the areas surrounding geosites determine the quality of the landscape, which is of great importance in tourism. The natural environment and the landscape are essential resources for tourism development [41]. In this study, in addition to facilities' values, landscape values that are important for tourism development were also taken into account.

## 2. Materials and Methods

### 2.1. Study Area

The study area is located in the Fann Mountains in the Zarafshan Range (NW Tajikistan), which is in the western part of Pamiro-Alay. The Zarafshan Range is approximately 350 km long, and the average width is 50 km [42]. The Fann Mountains are the highest part of the Zarafshan Range, with average altitudes reaching about 4100 m and the highest peak being Chimtarga Peak (5489 m). The range is characterised by a typical high-altitude relief, which is associated with their formation during the Alpine orogenesis [43]. Elements of the relief are its U-shaped valleys, glacial cirques, steep slopes, and numerous landslide

cones. The highest parts of the mountains are covered with glaciers, with glacial rivers in the valleys and glacial circuses filling the lakes of Kuli Kalon, Alouddin, Dushakha, and Iskandarkul. The Fann Mountains are built mainly of rocks of Paleozoic (Carboniferous, Devonian, Silurian) and Mesozoic (mainly Cretaceous) origin. Younger Quaternary deposits fill the bottom of the valleys. Rocks are made up of sandstones, quartzites, gneisses, limestones, marls, and dolomites, as well as clays and loess [44]. There are also hard coal deposits, which are an exploited rock raw material.

The area is characterised by climate and vegetation altitudinal zonation. In the valley bottoms and on the lower parts of the slopes (1700–2000 m), the warmest month is July, with an average temperature of about 19 °C, while the coldest is January, with an average of about 4 °C [43]. The mean yearly precipitation is 400–500 mm on peaks and slopes at altitudes of about 3000–3400 m a.s.l. In the basins and deep valleys (altitudes of 2200–2500 m a.s.l.), precipitation drops to 250–300 mm. Most precipitation occurs in the spring season (60–70 mm in April or May), while winter features a low level of snowfall. The valleys are covered with grass formations and juniper forests with *Juniperus seravschanica* and *Juniperus semiglobosa* [45]. In the higher parts (3000–3400 m), the average temperature in July is 10 °C, and in January, −10 °C. These areas are overgrown with rare juniper forests and torn vegetation from *Juniperus turkiestanica*, *Festuca ulcate*, and *Onobrychis echidna*, with *Oxytropis savellanica* in higher parts. Above the snow line (about 4000 m), average annual temperatures reach about −5 °C, while at Chimtarga, they reach −15 °C [42,46].

The Fann Mountains are protected as a national park. In addition, part of it has appeared on the UNESCO Tentative List of natural objects since 2006. This is due to its high biodiversity of flora and fauna and the high-mountain landscape with its characteristic glacial relief [47]. Species of plants from the Red Book of the Republic of Tajikistan and animals such as snow leopard, bearded eagle, golden eagle, turkestanic desert falcon, and asian moufflon all live in the park [47]. Due to its high ecological and landscape value, it is a region highly exposed to human influence. On the other hand, the same values make tourism attractive. In the Development Program for 2009–2019 for Tajikistan, mountain tourism and promotion of regions based on natural values were among the most important aims. Additionally, the document emphasised the development of ecotourism in the area of national parks [20]. Tourism in Tajikistan has been operating since 1962 based on camping and mountain tourism, and the Fann Mountains area was and is still one of the most visited. The main types of tourism in the region are adventure, mountain, and nature-based [20].

For detailed investigation of the tourist values of hydrogeosites, the authors selected 12 of them (Figure 1).

### 2.2. Materials and Methods

The value of hydrogeosites was assessed using various materials. The main group of materials includes field notes, field sketches, field measurement results, and photographic documentation. These materials were collected during field studies in June 2019. The last group is comprised of cartographic materials: Shuttle Radar Topography Mission (SRTM), an orthophotomap of Tajikistan, and a topographic map. The last group includes information about tourist offers, which comes from internet research on travel agencies and tour organisers in the area of the Fann Mountains (Table 2).

To assess the hydrogeosites, we used the modified method proposed by P. Pereira et al. [37] (Table 1). The modification concerns the criteria of accessibility (Ac) of hydrogeosites, which were adapted to the conditions of the Fann Mountains. We used this method to assess the internal values of the hydrogeosites that are connected to the features and locations of the geosites. In our study, we extended the assessment by adding criteria of external values that are connected to the surrounding landscape.

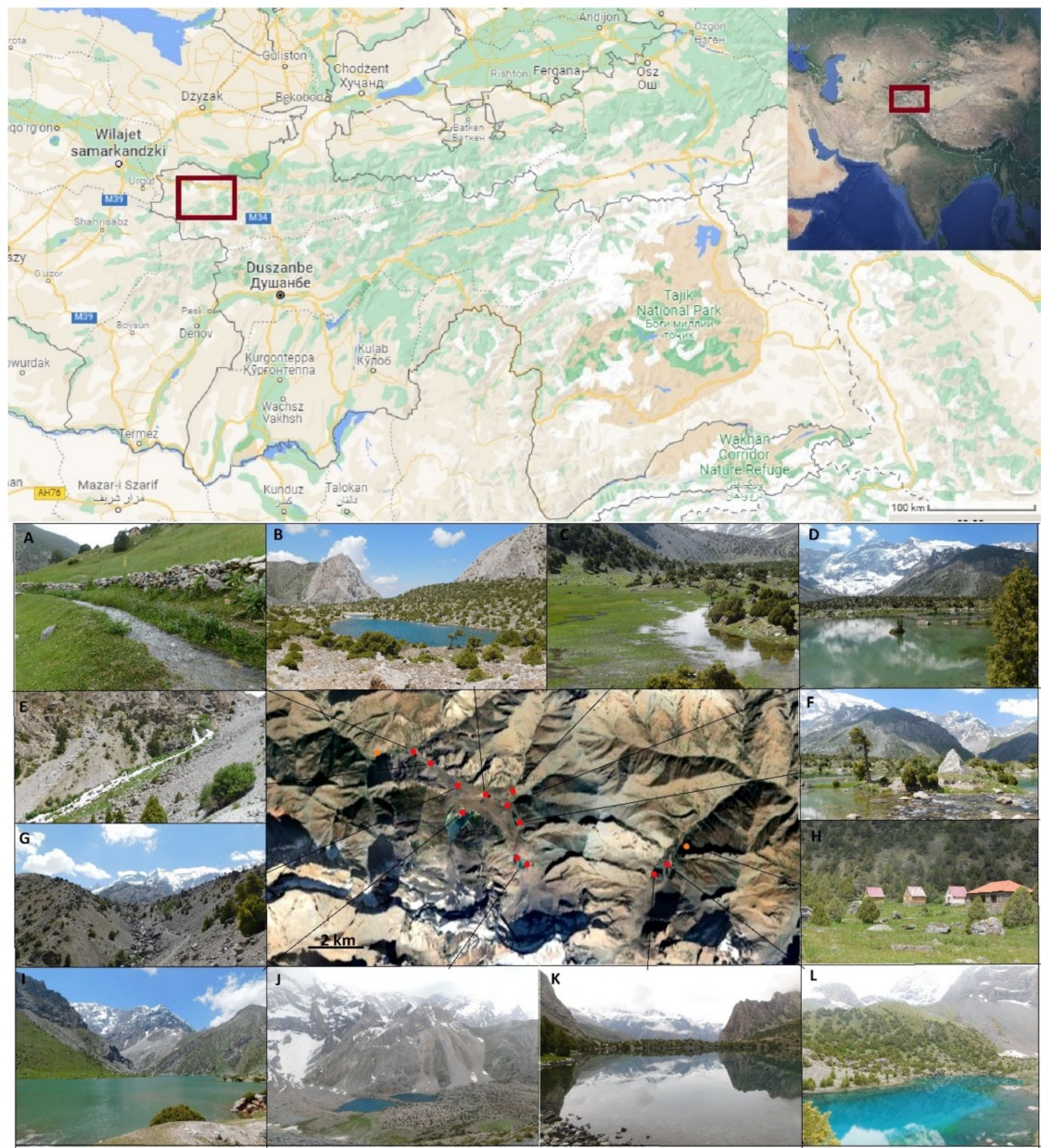

**Figure 1.** Location of hydrogeosites. Explanations: (**A**)—Irrigation Channels in the Urech Valley, (**B**)—Kuli Kalon, (**C**)—Swampland in the Kuli Kalon Plateau, (**D**)—Kuli Kalon, (**E**)—Waterfall in the Urech Valley, (**F**)—Bibijonat Lake, (**G**)—Exsurgent in the Urech Valley, (**H**)—Vertikal Alpine Base, (**I**)—Great Kuli Kalon Lake, (**J**)—Dushakha Lake 1 and 2, (**K**)—Allaudin Lake 2, (**L**)—Allaudin Lake 1. Source: own elaboration based on Google Maps and Google Earth.

**Table 1.** Numerical assessment of the hydrogeosites indicator by Pereira et al. [37].

| Criterion | Value |
|:---:|:---:|
| **A. SCIENTIFIC VALUE (ScV) (maximum 5.5)** | |
| **Rareness in relation to the area (Ra)** | |
| It is not one of the most important 5 | 0 |
| It is not one of the most important 3 | 0.25 |
| One of the most important 3 | 0.5 |
| The most important | 0.75 |
| The only occurrence | 1.0 |

**Table 1.** *Cont.*

| Criterion | Value |
|---|---|
| **In Integrity/Intactness (In)** | |
| Highly damaged as a result of human activities | 0 |
| Damaged as a result of natural processes | 0.25 |
| Damaged but preserving essential geomorphological features | 0.5 |
| Slightly damaged but still maintaining the essential geomorphological features | 0.75 |
| No visible damage | 1.0 |
| **Representativeness of geomorphological processes and pedagogical interest (Rp)** | |
| Low representativeness and without pedagogical interest | 0 |
| With some representativeness but with low pedagogical interest | 0.33 |
| Good example of processes but hard to explain to non-experts | 0.67 |
| Good example of processes and/or a good pedagogical resource | 1 |
| **Number of interesting geomorphological features (diversity) (Dv)** | |
| 1 | 0 |
| 2 | 0.33 |
| 3 | 0.67 |
| More than 3 | 1 |
| **Other geological features with heritage value (Hy)** | |
| Absence of other geological features | 0 |
| Other geological features but without relation to geomorphology | 0.17 |
| Other geological features with relation to geomorphology | 0.33 |
| Occurrence of other geosite(s) | 0.5 |
| **Scientific knowledge on geomorphological issues (Kn)** | |
| None | 0 |
| Medium: presentations, national papers | 0.25 |
| High: international papers, thesis | 0.5 |
| **Rareness at national level (Ra)** | |
| More than 5 occurrences | 0 |
| Between 3 and 5 occurrences | 0.17 |
| 2 occurrences | 0.33 |
| The only occurrence | 0.5 |
| **ScV Scientific value (Ra + In + Rp + Dv + Ge + Kn + Rn)** | |
| **B. ADDITIONAL VALUES (AdV) (maximum 4.5)** | |
| **Cultural value** | Cult |
| Without cultural features or with cultural features damaging the site | 0 |
| Cultural features with no connection to landforms | 0.25 |
| Relevant cultural features with no connection to landforms | 0.5 |
| Immaterial cultural features related to landforms | 0.75 |
| Material cultural features related to landforms | 1 |
| Relevant material cultural features related to landforms | 1.25 |
| Anthropic landform with high cultural relevance | 1.5 |

**Table 1.** *Cont.*

| Criterion | | Value |
|---|---|---|
| **Aesthetic value** | | Aest |
| Subjective value. Aspects to be considered: visual singularity of landforms; panoramic quality; objects and colour diversity and combination; presence of water and vegetation; absence of human-induced deterioration; proximity to the observed features. | Low | 0–0.5 |
| | Medium | 0.5–1 |
| | high | 1–1.5 |
| **Ecological value** | | Ecol |
| Without relation to biological features | | 0 |
| Occurrence of interesting fauna and/or flora | | 0.38 |
| One of the best places to observe interesting fauna and/or flora | | 0.75 |
| Geomorphological features are important for ecosystem(s) | | 1.12 |
| Geomorphological features are crucial for the ecosystem(s) | | 1.5 |
| **AdV Additional values (Cult + Aest + Ecol)** | | |
| **C. USE VALUE (UsV)** (maximum 7.0) | | |
| **Accessibility** | | Ac |
| Very difficult, only with special equipment | | 0 |
| Only by 4-wheel-drive vehicle and more than 500 metres by footpath | | 0.21 |
| By car and more than 500 metres by footpath | | 0.43 |
| By car and less than 500 metres by footpath | | 0.64 |
| By 4-wheel-drive vehicle and less than 100 metres by footpath | | 0.86 |
| By car to camp base and less than 50 metres by footpath | | 1.07 |
| By car on local roads and less than 1000 metres by footpath | | 1.29 |
| By car on national roads and less than 50 metres by footpath | | 1.5 |
| **Visibility** | | Vi |
| Very difficult to view or not visible at all | | 0 |
| Can only be viewed using special equipment (e.g., artificial light, ropes) | | 0.3 |
| Limited by trees or lower vegetation | | 0.6 |
| Good but need to move around for a complete observation | | 0.9 |
| Good for all relevant geomorphological features | | 1.2 |
| Excellent for all relevant geomorphological features | | 1.5 |
| **Present use of the geomorphological interest** | | Gu |
| Without promotion and not being used | | 0 |
| Without promotion but being used | | 0.33 |
| Promoted/used as a landscape site | | 0.67 |
| Promoted/used as a geomorphosite or geosite | | 1 |
| **Present use of other natural and cultural interests** | | Ou |
| Without other interests, promotion, and use | | 0 |
| With other interests but without promotion and use | | 0.33 |
| With other interests and their promotion, but without other use | | 0.67 |
| With other interests, with promotion and use | | 1 |

**Table 1.** *Cont.*

| Criterion | Value |
|---|---|
| **Legal protection and use limitations** | Lp |
| With total protection and prohibitive use | 0 |
| With protection, with use restriction | 0.33 |
| Without protection and without use restriction | 0.67 |
| With protection but without use restriction or with very low use restriction | 1 |
| **Equipment and support services** | Eq |
| Hostelry and support services are more than 25 km away | 0 |
| Hostelry and support services are between 10 and 25 km away | 0.25 |
| Hostelry and support services are between 5 and 10 km away | 0.5 |
| Hostelry or support services are less than 5 km away | 0.75 |
| Hostelry and support services are less than 5 km away | 1 |
| **UsV Use value (Ac + Vi + Gu + Ou + Lp + Eq)** | |

The analysis of the value of the hydrogeosites for tourist purposes was carried out in three stages (Figure 2). The first stage was the inventory of available hydrogeosites in the Fann Mountains. For this purpose, the authors carried out a field study in June 2019 and Internet research of available tourist offers in the Fann Mountains. Since water plays an important role in the studied area, natural, cultural, and economic hydrogeosites were taken into account as the geosites [48]. During the field study, the authors identified potential hydrogeosites in the study area and assessed them using the adopted criteria. The selection criteria were field availability, hydrological nature of the site, and potential tourist values determined based on available tourist offers. The hydrogeosites that fulfilled these requirements were selected as objects of investigation. Twelve hydrogeosites that are both natural and anthropogenic hydrological objects were selected for further analysis. They are all located along the main tourist trail from the climbing base in Artouch to the Vertikal alpine base.

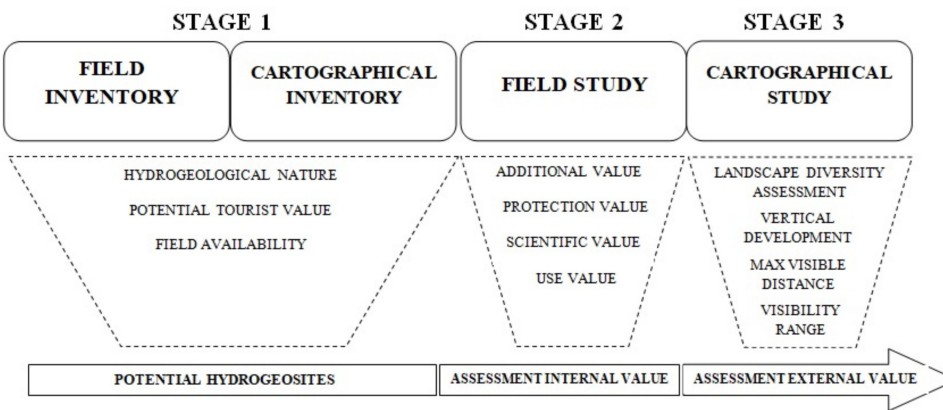

**Figure 2.** Schematic diagram of the research procedure.

The second stage of hydrogeosite analysis was the assessment of their internal value. The amount obtained allowed for positions ranging from the least valuable under the selected criteria to the highest internal attractiveness. The higher the sum of points, the more attractive the hydrogeosite.

The third stage of the study was an external evaluation of the hydrogeosites. To date, assessments have been carried out only taking into account the internal criteria of the sites themselves. Their external conditions as landscape values, which significantly affect their

tourist attractiveness, have not been assessed. To assess their external value, the following criteria were applied:

- analysis of the visibility range (VR)—the area visible from particular points;
- vertical development (MVD), as the difference between the maximal and minimal altitude visible from particular points (relative height);
- maximum visibility range (VD), as the horizontal distance (in kilometers) at which a large dark object can just be seen against the horizon sky in daylight; and
- landscape diversity (LD) assessment—the number of land cover types, such as forests, alpine halls, and rock surfaces, visible from particular points.

The higher the sum of points obtained, the higher the tourist value of the investigated hydrogeosite.

## 3. Results

### 3.1. Analysis of Available Tourist Offers

A review of available tourist offers indicates that the main types of tourism are hiking (nine offers), mountain trekking (nine offers), and climbing (three offers). The shortest offers last 4 days, and the longest 16 days. Considering expedition organisers, foreign organisers dominate (six offers), and only three offers were prepared by tourist offices from Tajikistan. Most of the offers are European (four), and two come from neighbouring countries (Kyrgyzstan and Uzbekistan). Expedition prices vary and depend on the organiser's country and length of stay. However, it can be seen that offers from Europe (United Kingdom, Switzerland) are almost twice as expensive as local offers. Artuch Base Camp is used as a place to stay in eight expeditions, while Vertikal appears in one offer. The hydrogeosites visited during expeditions are lakes or lake complexes: Kuli Kalon Lakes, Alouddin Lakes (all offers—nine), Mutnoye Lake (six offers), Bibijanat Lake (five offers), Iskandarkul Lake, Chukurak Lake (three offers), Alouddin Lakes (two offers), Mazguzor Lake, Dushakha Lakes (one offer). Passes are also important points of interest: Alouddin Pass (five trips), Chmitarga Pass (three), Munora Pass, and Mirali Pass. For climbing expeditions, the goal is to reach the Chimtarga and Energia peaks (Table 2).

Currently, one of the most developed companies serving mountain tourism in Tajikistan in the area of the Fann Mountains (District of Seven Lakes, Kulikalon, Kuli Alloudin, Iskandarkul) is Artuch Travel. The company owns Mountain Base Artuch (Figure 3), founded in 1971 and located at a height of 2200 m a.s.l. in the Urech valley in the Fann Mountains. This territory was kolkhoz during the times of the Union of Soviet Socialist Republics. These were agricultural production collectives that consisted of the lands of the peasants. They were created in the process of collectivisation of agriculture. The Artuch Company provides services for tourists who want to stay in the camp for some time as well as backpackers who are just passing through. There are economy class and standard rooms, suites, and a campsite. Meals are prepared by a cosy cafe located in the territory.

During the times of the Union of Soviet Socialist Republics, there was an Alpine centre supported by the headquarters in Moscow. Because of its location on a former kolkhoz area, it was guarded. As a result, an archa forest (the name 'archa' means juniper in Tajiki) has survived (Figure 3b). It is a very crucial area because significant destruction of the archa forests in Central Asia was reported as early as the end of the nineteenth century and has continued until the present day [46]. It should be pointed out that the mountain ecosystems of Central Asia, including forest systems, are considered to be among the most valuable regions of the Earth in terms of biodiversity, referred to as 'hot spots of biodiversity' [49].

These forests are a matter of concern for eco-tourists from Western Europe (Germany, France, Great Britain). Hence, Mountain Base Artuch organises trekking for tourists from the abovementioned areas.

**Table 2.** Tourist offers in the Fann Mountains (source of information: accessed on 10 February 2019).

| Name of Offer | Point of Tour in Investigated Area (Hydroeosites) | Duration | Country of Origin of Offer/Organiser | Type of Tourism | Price | Source of Information |
|---|---|---|---|---|---|---|
| Tajikistan Trekking Tour | Artuch Camp, Kuli Kalon Lakes, Alouddin Lakes, Iskandar Kul | 11 days | Switzerland/Kalpak Travel | hiking, trekking | 1890 EUR | kalpak-travel.com |
| Fann Mountains Trek and Silk Road Cities | Atruch Base Camp, Kuli Kalon Lakes, Alouddin Lakes, Mutnoye Lake | 14 days (5-day trek) | United Kingdom/World Expeditions | hiking, trekking | 2270 EUR | worldexpeditions.com |
| Trekking Fann Mountains | Artuch Base Camp, Kuli Kalon Lakes, Bibijannat Lake, Alouddin Pass, Alouddin Lakes, Mutnoye Lake | 4-day trek | Kyrgyz/Visit Alay | hiking, trekking | no data | visitalay.com |
| Trekking in Fann Mountains | Atruch Base Camp, Chukurak Lake, Kuli Kalon Lakes, Bibijannat Lakes, Alouddin Lakes, Alouddin Pass, Mutnoye Lake, Vertikal Camp | 6-day trek | Tajikistan/Paramount Journey | hiking, trekking | no data | paramountjourney.com |
| Heartland of Fann Mountains | Artuch Base Camp, Kuli Kalon Lakes, Alouddin Lakes, Mutnoye Lake, Chimtarga Pass | 8 days | Uzbekistan/Kyrgyzstan/Russia/ Central Asia Travel | hiking, trekking, climbing | 1295 USD | centralasia-travel.com |
| Fann Mountains. Trekking in Fann Mountains | Lake Chukurak, Kuli Kalon Lakes, Bibijanat Lake, Dushakha Lakes, Alouddin Pass, Alouddin Lakes, Mutnoye Lake, Chimtarga Pass, Allo Lakes, Munora Pass, Mazguzor Lake | 12–14 days | Russia/ClimberCA Mountaineering Asia | trekking, hiking, climbing | no data | climberca.com |
| Trekking in Fann Mountains | Artuch Base Camp, Bibijanat Lake, Kuli Kalon Lakes, Alouddin Lakes, Iskandarkul Lake | 8 days | Tajikistan/Viator | trekking, hiking | 1443 USD | viator.com |
| Climb Chimtarga (5489 m) and trek the Fann Mountains, Tajikistan | Artuch, Kuli Kalon Lakes, Chukurak Lakes, Alouddin Pass, Alouddin Lakes, Mutnoye Lake, Energia, Mirali Pass, Chmitarga Pass, Chimtarga, Allo Lake | 16 days | United Kingdom/Remote Corner Adventures | trekking, hiking, climbing | 2099 USD | remote-corner.com |
| Trekking in Fann Mountains | Artuch, Bibijanat Lake, Kuli Kalon Lakes, Alouddin Pass, Alouddin Lakes, Iskanderkul Lake, | 8 days | Tajikistan/Indy Guide | trekking, hiking | 699 USD | indy-guide.com |
| Tajikistan Trekking Tour | Artuch Camp, Kuli Kalon Lakes, Alouddin Lakes, Iskandar Kul | 11 days | Switzerland/Kalpak Travel | hiking, trekking | 1890 EUR | kalpak-travel.com |
| Fann Mountains Trek and Silk Road Cities | Atruch Base Camp, Kuli Kalon Lakes, Alouddin Lakes, Mutnoye Lake | 14 days (5-day trek) | United Kingdom/ World Expeditions | hiking, trekking | 2270 EUR | worldexpeditions.com |
| Trekking Fann Mountains | Artuch Base Camp, Kuli Kalon Lakes, Bibijannat Lake, Alouddin Pass, Alouddin Lakes, Mutnoye Lake | 4-day trek | Kyrgyz/Visit Alay | hiking, trekking | no data | visitalay.com |
| Trekking in Fann Mountains | Atruch Base Camp, Chukurak Lake, Kuli Kalon Lakes, Bibijannat Lakes, Alouddin Lakes, Alouddin Pass, Mutnoye Lake, Vertikal Camp | 6-day trek | Tajikistan/Paramount Journey | hiking, trekking | no data | paramountjourney.com |
| Heartland of Fann Mountains | Artuch Base Camp, Kuli Kalon Lakes, Alouddin Lakes, Mutnoye Lake, Chimtarga Pass | 8 days | Uzbekistan/Kyrgyzstan/Russia/ Central Asia Travel | hiking, trekking, climbing | 1295 USD | centralasia-travel.com |

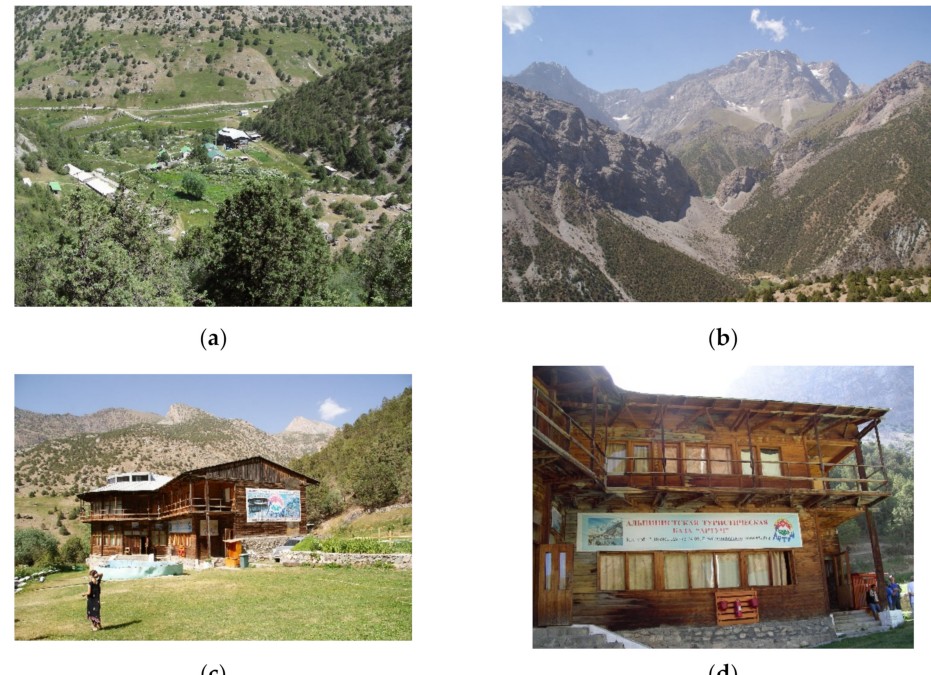

**Figure 3.** Artuch Camp and its surroundings: (**a**) a view of the camp from the surrounding peaks; (**b**) juniper forests surrounding the camp; (**c,d**) the main building of the camp.

*3.2. Internal Value Analysis*

Of the 12 analysed objects (Table 3), Great Kuli Kalon Lake received the highest total value (12.15 points) (Figure 4). Aloudin Lake 1 (10.65), Swampland in the Kuli Kalon Plateau (11.19), and Lake Kuli Kalon 2 (10.06) also obtained high values. Other objects did not achieve half of the possible points, so they have ratings lower than 10 points. Lake Alouddin 2 received the lowest value of 9.1 points.

The highest scientific value (Figure 4a) was obtained by objects in the Urech Valley: the Exsurgent (3.75) and the Waterfall (3.41). Bibijanat Lake and Lake Kuli Kalon 1 (both 1.75) have the lowest scientific value. All the evaluated positions did not receive points for two criteria—scientific knowledge of hydrogeosite issues (KN) and rareness at the national level (RN). The degree of scientific recognition of the analysed objects is poor; therefore, all received 0 points. None of the evaluated hydrosites is rare in Tajikistan; therefore, all of them received 0 points. In additional value (Figure 4b), the Swampland in the Kuli Kalon Plateau was rated the highest (4.02), and the Exsurgent in the Urech Valley received the least points (1.13). All analysed sites received a very low use value (Figure 4c), and only two of them achieved 50% of the maximum points. These are Great Kuli Kalon Lake (3.5) and the Irrigation Channels in the Urech Valley (3.82). The Dushakha Lakes (1.44) have the lowest use value. Two criteria stand out in the assessment—current use of the hydrological interest (Hu) and current use of other natural and cultural interests (Ou). In both criteria, only Great Kuli Kalon Lake received points (0.67) as a place promoted because of its landscape attractiveness. In addition, considering the legal protection and use limitations (LP) criterion, the Irrigation Channels were the only site to receive a higher value of points due to the lack of protection and restrictions on their use (Figure 4d). The highest protection value was obtained by the Exsurgent in the Urech Valley (3), while the lowest was obtained by the Irrigation Channels, also in the Urech Valley (0.5).

Great Kuli Kalon Lake is the only hydrogeosite that has all the utility values. Although this site did not obtain the maximum points in the other categories, it obtained the maximal value in this criterion. The Irrigation Channels are another distinctive object, being the only anthropogenic object. For this reason, they obtained the lowest value in terms of the degree of protection. However, due to their availability and agricultural suitability, they

have the highest utility value. Among the indicators, it is worth paying attention to the parameters Ac and Eq, which refer to the availability of hydrogeosites and their distance from accommodation bases, respectively. The points values of the evaluated objects are arranged in spatial regularity—the further from the accommodation base, the lower the score. So, it turns out that the least available hydrogeosites are the Dushakha Lakes.

**Table 3.** Internal assessment of hydrogeosites.

| Assessment Criteria/Hydrogeosites | Alouddin Lake 1 | Alouddin Lake 2 | Dushakha Lake 1 | Dushakha Lake 2 | Bibijonat Lake | Swampland in Kuli Kalon Plateau | Kuli Kalon Lake 1 | Kuli Kalin Lake 2 | Great Kuli Kalon Lake | Exsurgent in Urech Valley | Waterfall in Urech Valley | Irrigation Channels in Urech Valley |
|---|---|---|---|---|---|---|---|---|---|---|---|---|
| **ScV** | | | | | | | | | | | | |
| RA | 0 | 0 | 0 | 0 | 0 | 0 | 0 | 0 | 0 | 0.75 | 0.75 | 0.75 |
| IN | 1 | 1 | 1 | 1 | 0.75 | 1 | 0.75 | 1 | 1 | 1 | 1 | 0.5 |
| RP | 1 | 1 | 1 | 1 | 0.67 | 1 | 0.67 | 0.67 | 1 | 1 | 1 | 1 |
| DV | 0 | 0 | 0 | 0 | 0 | 0.33 | 0 | 0 | 0 | 0.67 | 0.33 | 0.33 |
| HY | 0.33 | 0.33 | 0.33 | 0.33 | 0.33 | 0.33 | 0.33 | 0.33 | 0.5 | 0.33 | 0.33 | 0 |
| KN | 0 | 0 | 0 | 0 | 0 | 0 | 0 | 0 | 0 | 0 | 0 | 0 |
| RN | 0 | 0 | 0 | 0 | 0 | 0 | 0 | 0 | 0 | 0 | 0 | 0 |
| Sum of ScV | 2.33 | 2.33 | 2.33 | 2.33 | 1.75 | 2.66 | 1.75 | 2 | 2.5 | 3.75 | 3.41 | 2.58 |
| **AdV** | | | | | | | | | | | | |
| CULT | 1.5 | 0 | 1.5 | 1.5 | 1.5 | 1.5 | 1.5 | 1.5 | 1.5 | 0 | 0 | 1.5 |
| AEST | 1.35 | 1.25 | 1.25 | 1.25 | 1.15 | 1.4 | 1.15 | 1.15 | 1.4 | 0.75 | 1.15 | 1 |
| ECOL | 0.75 | 0.75 | 0.75 | 0.75 | 0.75 | 1.12 | 0.75 | 0.75 | 0.75 | 0.38 | 0 | 0.38 |
| Sum of AdV | 3.55 | 2 | 3.5 | 3.5 | 3.40 | 4.02 | 3.4 | 3.4 | 3.65 | 1.13 | 1.15 | 2.88 |
| **UsV** | | | | | | | | | | | | |
| Ac | 0.29 | 0.29 | 0.21 | 0.21 | 0.43 | 0.43 | 0.43 | 0.43 | 0.43 | 0.43 | 0.86 | 1.5 |
| Vi | 0.9 | 0.9 | 0.9 | 0.9 | 0.9 | 1.5 | 0.9 | 0.9 | 0.9 | 0.6 | 1.2 | 0.9 |
| Hu | 0 | 0 | 0 | 0 | 0 | 0 | 0 | 0 | 0.67 | 0 | 0 | 0 |
| Ou | 0 | 0 | 0 | 0 | 0 | 0 | 0 | 0 | 0.67 | 0 | 0 | 0 |
| Lp | 0.33 | 0.33 | 0.33 | 0.33 | 0.33 | 0.33 | 0.33 | 0.33 | 0.33 | 0.33 | 0.33 | 0.67 |
| Eq | 0.75 | 0.75 | 0 | 0 | 0.25 | 0.25 | 0.25 | 0.5 | 0.5 | 0.5 | 0.75 | 0.75 |
| Sum of UsV | 2.27 | 2.27 | 1.44 | 1.44 | 1.91 | 2.51 | 1.91 | 2.16 | 3.5 | 1.86 | 3.14 | 3.82 |
| **VPr** | | | | | | | | | | | | |
| IN | 1 | 1 | 1 | 1 | 0.75 | 1 | 0.75 | 1 | 1 | 1 | 0.5 | 0.5 |
| VU | 1.5 | 1.5 | 1.5 | 1.5 | 1.5 | 1 | 1.5 | 1.5 | 1.5 | 2 | 2 | 0 |
| Sum of VPr | 2.5 | 2.5 | 2.5 | 2.5 | 2.25 | 2 | 2.25 | 2.5 | 2.5 | 3 | 2.5 | 0.5 |
| **Sum of ScV + AdV + UsV + VPr** | **10.65** | **9.1** | **9.77** | **9.77** | **9.31** | **11.19** | **9.31** | **10.06** | **12.15** | **9.74** | **10.2** | **9.78** |

### 3.3. External Value Analysis

The highest visibility range was found in the sites on Alouddin Lake 1 and 2 (48.3 km$^2$ and 47.5 km$^2$, respectively) (Table 4 and Figure 5). The smallest visibility range is characterised by sites in the Urech Valley—the Exsurgent (4.2 km$^2$) and the Waterfall (9 km$^2$). The maximal visible distance of 31.3 km was determined from Lake Kuli Kalon 1. The lowest value of this indicator was found in Lake Bibijonat, where it is 6.6 km. The largest vertical development was measured from Lake Kuli Kalon 2 (2451 m), from where the highest peak

of the Fann Mountains—Chimtarga—is visible (5489 m), while the smallest was on the site of the Exsurgent in the Urech Valley (568 m). The highest value of landscape diversity was found near the Swampland in the Kuli Kalon Plateau and Irrigation Channels in the Urech Valley. In these places, visible land cover is the most diverse with as many as seven land cover types being found (Table 4).

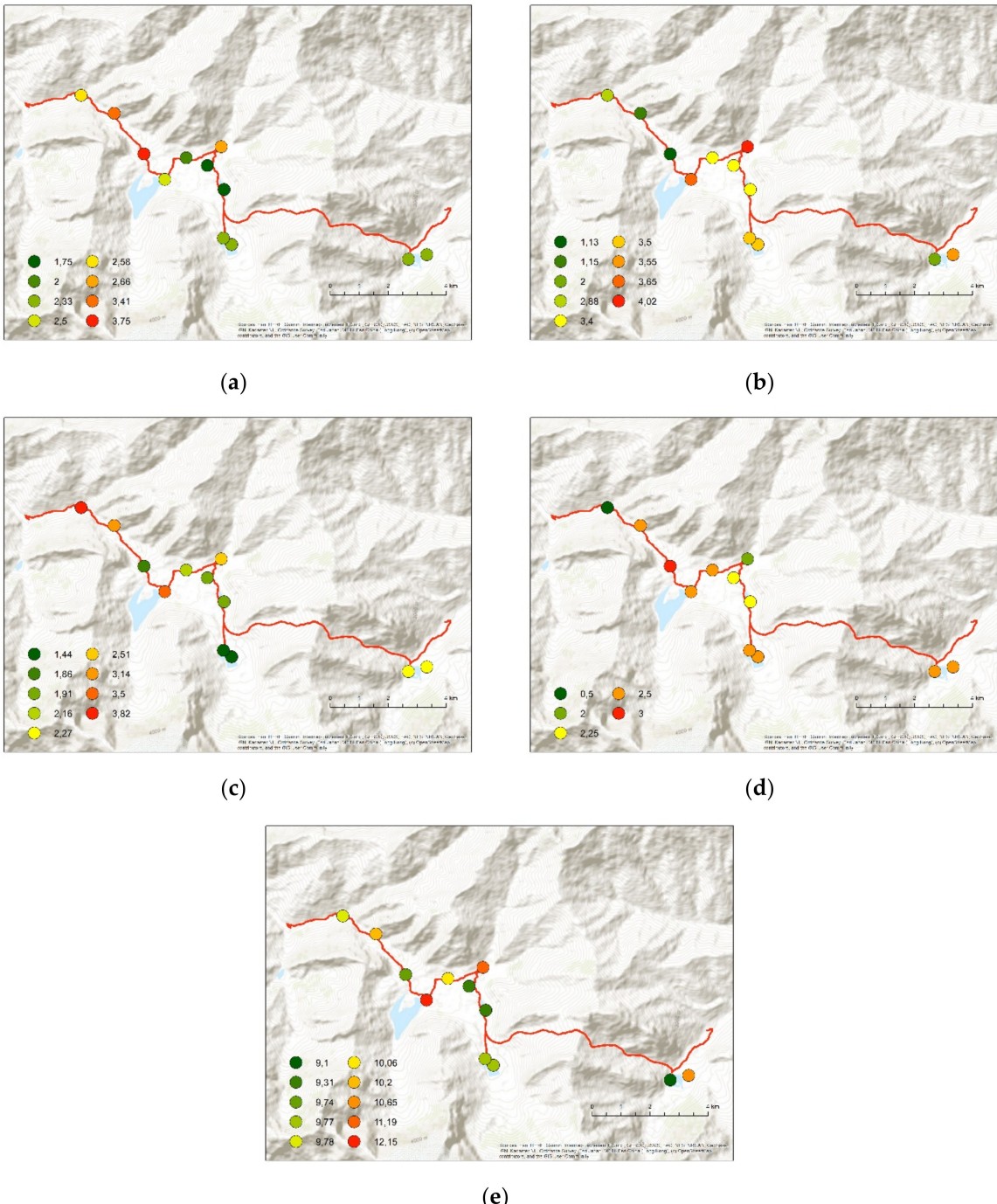

**Figure 4.** Internal values of the analysed hydrogeosites: (**a**) scientific value; (**b**) additional values; (**c**) use values; (**d**) protection value; (**e**) total. Source: own elaboration based on World Topographic Map, 19 February 2012, arcgis.com (accessed on 2 November 2021).

Table 4. External assessment of hydrogeosites.

| Assessment Criteria/Hydrogeosites | Alouddin Lake 1 | Alouddin Lake 2 | Dushakha Lake 1 | Dushakha Lake 2 | Bibijonat Lake | Swampland in Kuli Kalon Plateau | Kuli Kalon Lake 1 | Kulikalin Lake 2 | Great Kuli Kalon Lake | Exsurgent in Urech Valley | Waterfall in Urech Valley | Irrigation Channels in Urech Valley |
|---|---|---|---|---|---|---|---|---|---|---|---|---|
| | | | | | | SuV | | | | | | |
| VR (km$^2$) | 48.3 | 47.5 | 24.3 | 27 | 17.7 | 35.3 | 36.5 | 32.9 | 27.6 | 4.2 | 9 | 13.4 |
| MVD (km) | 16.5 | 18.7 | 8.4 | 8.3 | 6.6 | 11 | 31.3 | 13.7 | 7.7 | 12 | 20.1 | 19.4 |
| VD (m) | 2225 | 2380 | 2219 | 2059 | 2416 | 2434 | 2438 | 2451 | 1910 | 568 | 1240 | 2178 |
| LD | 5 | 6 | 5 | 5 | 5 | 7 | 5 | 5 | 6 | 3 | 4 | 7 |

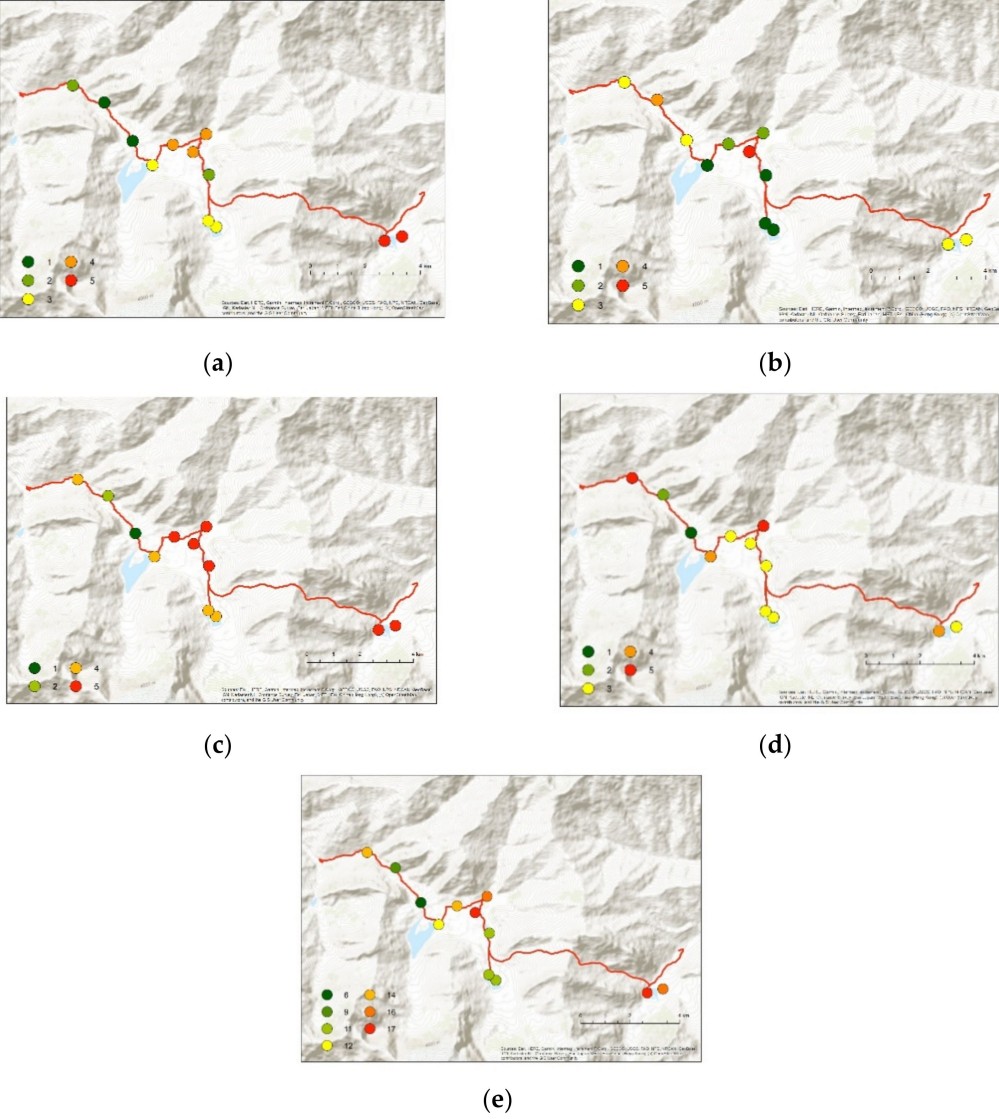

(a)

(b)

(c)

(d)

(e)

**Figure 5.** External values of the analysed hydrogeosites: (**a**) visibility range; (**b**) visibility development; (**c**) maximal visible distance; (**d**) landscape diversity; (**e**) total. Source: own elaboration based on World Topographic Map, 19 February 2012, arcgis.com (accessed on 2 November 2021).

The obtained results were changed into five categories (from 1 for the lowest external value to 5 for the highest). Then, they were added together (Table 5). The highest value of the environment was obtained by Lake Kuli Kalon 1 and Alouddin 2 (17). The surroundings of Lake Alouddin 1 and the Swampland in the Kuli Kalon Plateau (16 each) were also highly rated. The lowest value of the surroundings was found in the Urech Valley sites where the Exsurgent obtained six points and the Waterfall nine points (Table 5). The relief most influences the assessment of the external values because sites in open spaces obtained higher values than sites in partially closed forms for the surrounding area (valleys).

**Table 5.** Assessment of the surroundings of hydrogeosites.

| Assessment Criteria/Hydrogeosites | Alouddin Lake 1 | Alouddin Lake 2 | Dushakha Lake 1 | Dushakha Lake 2 | Bibijonat Lake | Swampland in Kuli Kalon Plateau | Kuli Kalon Lake 1 | Kulikalin Lake 2 | Great Kuli Kalon Lake | Exsurgent in Urech Valley | Waterfall in Urech Valley | Irrigation Channels in Urech Valley |
|---|---|---|---|---|---|---|---|---|---|---|---|---|
| | | | | | | SuV | | | | | | |
| VR | 5 | 5 | 3 | 3 | 2 | 4 | 4 | 4 | 3 | 1 | 1 | 2 |
| MVD | 3 | 3 | 1 | 1 | 1 | 2 | 5 | 2 | 1 | 3 | 4 | 3 |
| VD | 5 | 5 | 4 | 4 | 5 | 5 | 5 | 5 | 4 | 1 | 2 | 4 |
| LD | 3 | 4 | 3 | 3 | 3 | 5 | 3 | 3 | 4 | 1 | 2 | 5 |
| **Sum of SuV** | **16** | **17** | **11** | **11** | **11** | **16** | **17** | **14** | **12** | **6** | **9** | **14** |

## 4. Discussion

In this paper, we analysed the touristic attractiveness of hydrogeosites in the Fann Mountains, taking into account the current tourism development stage in this area. Firstly, we analysed the available tourist offers to assess the stage of tourism development, and then we assessed the attractiveness of the hydrogeosites taking into account their internal and external values.

The current offers suggest that the Fann Mountains are in the exploration stage of the Butler Model [21]. A small number of tourists are attracted by primary tourist attractions, both natural and cultural. There are no secondary tourism attractions, and tourism has no economic or social significance to local residents. The current tourism stage gives us a chance to implement principles of sustainable development before tourism develops in an uncontrolled way [50]. Unfortunately, mountain tourism in developing countries is mostly characterised by haphazard planning and a lack of environmental standards and monitoring. Some of the problems of mountain tourism in developing countries are competition between small-scale local operations and large international chains, alienation of local residents due to many visitors, and environmental damage [51]. There are dangers in promoting mountain destinations for tourism, especially if there is no strategic focus on the type and intensity of activities to be promoted. Hence, the recognition of natural and cultural resources of tourist attractions is necessary.

As far as internal values are concerned, the analysed objects obtained a low scientific value, resulting from the low level of scientific recognition and common occurrence in Tajikistan. The maximum values are connected to integrity and representativeness. This area is characterised by extensive human use of water resources.

The high cultural value of the analysed objects is connected to human activity in the Kuli Kalon Basin. This area is utilised for traditional mountain grazing and tourism. In

turn, the highest aesthetic value of the objects themselves is found in all the lakes. Lakes are among the most aesthetic elements of the environment and are crucial in tourism [52]. High mountain lakes are called mountain pearls due to their turquoise water. The ecological value of the analysed hydrogeosites is high. They are a good place for observing animal and plant species. Hence, there is the potential to develop wildlife tourism. The Fann Mountains, similarly to the entire area of Tajikistan, show high floristic diversity [53]. This diversity results from the variations in the relief, microrelief, climate, microclimate, and topoclimate reflecting their altitudinal zonation. This has contributed to plant species developing that are specific to this geographical region alone, i.e., endemic species. Different species are associated with particular ecosystem types. The largest number of endemic species occurs between the altitudes of 1400 and 4000 m a.s.l. The endemic species are most common in the phytocoenoses involving juniper species, so it can be said that juniper forests represent a potential habitat for these taxa [45].

All objects obtained a low use value connected to the poor accessibility and lack of contemporary use as geosites. This is connected to the fact that geotourism is a new type of tourism that is still developing. Hence, this area is suitable for trekking, which is typical of high mountain areas [54]. The maximal use value was obtained by objects located near mountain base camps.

The sum of points obtained for the individual criteria varied greatly, which confirms the complexity of the landscape. Mountain landscapes in particular are characterised by considerable diversity, which results in a great variety of tourist values of different mountain locations [55]. In tourism, in addition to the value of tourist attractions, their surroundings are also important, which can be understood as landscape values. Their importance was pointed out by M. Rogowski [56], who emphasised that the observation of a closer or farther view is one of the elements of mountain tourism. A similar opinion is held by Jiménez-García et al. [57], who claim that landscape is a factor that attracts and develops tourism. Landscape values can be evaluated from different perspectives, but particular attention should be paid to the various elements that build the landscape and its horizontal and vertical development [58]. Scenic values often determine the choice of a given place as a tourist destination. This is why they should be taken into account in valorisation and the assessment of tourist attractiveness. The method used by P. Pereira [37] was mainly limited to the assessment of the geosites themselves. It must be emphasised that the landscape is very dynamic. Hence, its internal value should be assessed regularly. The direction and dynamics of landscape changes could be different in particular parts of the mountains [59].

The proposed method represents a holistic approach to assessing geosites. It takes both qualitative and quantitative aspects into account. Some of the criteria of assessment could be subjective. However, even during the quantification stage, it would seem to be impossible to avoid subjectivity. This is connected to the fact that the allocation of values for most criteria depends on the assessor's opinion. Nevertheless, the presented approach puts greater demands on the expertise of the assessor by including scientific and non-scientific criteria (such as additional values, potential use, and management) for judgement.

## 5. Conclusions

The study made it possible to assess the usefulness of hydrogeological sites for tourism purposes based on available data sources. However, it should be noted that there is a lack of detailed scientific studies on the selected hydrogeosites, which might cause the results to be underestimated. Therefore, these results cannot be compared to results from other areas that used the same assessment method. In the future, after the hydrogeological value of the sites has been scientifically recognised, the evaluation of their geotourism value should be repeated. It will then fully reveal the suitability of these sites for tourism development.

The conducted value assessments of the sites and their surroundings showed that, in both internal and external assessments, the highest values were achieved by lakes and wetlands. These are places that can be considered as vantage points. This confirms the

validity of assessing the surroundings for the purpose of selecting tourist attractions. It raises the objective value of the sites additionally by aesthetic values, which are very important from the point of view of tourists. Thus, the hydrogeological sites with the highest ratings have a dual function: (1) as educational sites for tourism; and (2) as vantage points. This means that the evaluation of the surroundings has a strong influence on the results obtained and the choice of hydrogeotourism attractions.

The analysis showed that the sites are kept in very good condition, which is due to their extensive use. The main factor limiting the deterioration of these objects is the orographic barrier, which makes the area difficult to access for other uses than tourism. This means that the area is a virgin and natural landscape. Moreover, due to the initial stage of tourism development, it is still an area where there is a chance to implement the principles of sustainable tourism. One such principle is to maintain the educational role of tourism and increase the awareness of visitors about the natural and cultural heritage of the Fann Mountains.

**Author Contributions:** Conceptualisation, K.P.-K. and M.S.; methodology, K.P.-K. and M.S.; software, M.S.; validation, O.R. and U.M.-P.; formal analysis, K.P.-K. and M.S.; investigation, K.P.-K., M.S. and O.R.; resources, K.P.-K., M.S. and O.R.; writing—original draft preparation, K.P.-K. and M.S.; writing—review and editing, O.R. and U.M.-P. All authors have read and agreed to the published version of the manuscript.

**Funding:** This research received no external funding.

**Institutional Review Board Statement:** Not applicable.

**Informed Consent Statement:** Not applicable.

**Data Availability Statement:** Publicly available datasets were analyzed in this study: https://srtm.csi.cgiar.org/ (accessed on 2 November 2021).

**Acknowledgments:** We gratefully acknowledge the two anonymous reviewers for their constructive comments.

**Conflicts of Interest:** The authors declare no conflict of interest.

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
