# Peer review of "The Assessment of Hydrogeosites in the Fann Mountains, Tajikistan as a Basis for Sustainable Tourism"

_resources, doi:10.3390/resources10120126_

Round 1

Reviewer 1 Report

The paper is well presented, English is correct, the is a good balance between the various parts of the paper, and the topic is interesting and fits with the scope of the journal.

Few minor recommendations are made in order to improve the final version (see attached manuscipt).

Author Response

Response to Reviewer 1

Reviewer 1: The paper is well presented, English is correct, the is a good balance between the various parts of the paper, and the topic is interesting and fits with the scope of the journal. Few minor recommendations are made in order to improve the final version (see attached manuscipt).

Author’s response: Thank you for the high rating of our article. The manuscript has been corrected according to the reviewers' comments.

Other Author’s comment: We sincerely thank all the reviewers for their constructive comments that helped to improve the article.

Reviewer 1: Usually numbers with less than five digits do not need commas.

Author’s comment: It was corrected.

Reviewer 1: Do the authors mean the Word Heritage List?  Please clarify it.

Author’s comment: It was corrected into: ‘The Fann Mountains are protected as a national park. In addition, part of it has appeared on the UNESCO Tentative List of natural objects since 2006’.  

Reviewer 1: Please add the explanation letters used in the caption to identify the hydrogeosites. This can help the reader to identify the geosites easily.

Author’s comment: We decided not to repeat the names of geosites. We removed it from the main text. It is included in figure caption. We hope that it is clear now.

Reviewer 1: What do you mean with "availability"? Please clarify it

Author’s comment: It was mistake. It should be accessilbility (Ac). It was corrected.

Reviewer 1: The table 1 citation is lacking. The table citation should on front of the table. Please check.

Author’s comment: It was corrected.

Reviewer 2 Report

The main contribution of the paper is to add the surroundings/setting of the geosites to an existing methodology.

Detailed points

Add Tajikistan to the title - "Fann Mountains, Tajikistan"

line 95 - omit "geosites"

line 109 - must include map(s) to show the study site and its location within Asia.

line 119 - replace "Dewon" with "Devonian"

lines 134-138 - use italics at species names

Table 1 - use bold type at (ScV), (Ra) and (In) as at (Rp), (AdV) etc.

lines 211-219 - add the letters VR, MVD, VD and LD to these descriptions. Perhaps give some examples of the "land cover types".

lines 248 and 255 - Explain "kolkhoz"

line 251 - replace "places for tents" with "campsite"

line 262 - add ' after biodiversity

line 303 - replace "criterium" with "criterion"

Author Response

Response to Reviewer 2

Reviewer 2: The main contribution of the paper is to add the surroundings/setting of the geosites to an existing methodology.

Reviewer 2: Detailed points: Add Tajikistan to the title - "Fann Mountains, Tajikistan"

Author’s response: Country name added to article title. Thank you for that comment.

Reviewer 2: line 95 - omit "geosites"

Author’s response: Corrected. One unnecessary word has been removed.

Reviewer 2: line 109 - must include map(s) to show the study site and its location within Asia.

Author’s response: Map with location within Asia and Tajikistan is added.

Reviewer 2: line 119 - replace "Dewon" with "Devonian"

Author’s response: Corrected into „Devonian”.

Reviewer 2: lines 134-138 - use italics at species names

Author’s response: The italic font is used for Latin names of plant species.

Reviewer 2: Table 1 - use bold type at (ScV), (Ra) and (In) as at (Rp), (AdV) etc.

Author’s response: Corrected errors in the table: incorrect numbering of criteria (now there is A, B, C instead of F, G, H) and the names of the main criteria are bold.

Reviewer 2: lines 211-219 - add the letters VR, MVD, VD and LD to these descriptions. Perhaps give some examples of the "land cover types".

Author’s response: Abbreviations have been added to the criteria descriptions, and examples of land cover types are also given.

Reviewer 2: lines 248 and 255 - Explain "kolkhoz"

Author’s response: The term „kolkhoz” was explained in the sentence where it was first used.

Reviewer 2: line 251 - replace "places for tents" with "campsite"

Author’s response: Replaced.

Reviewer 2: line 262 - add ' after biodiversity

Author’s response: Added.

Reviewer 2: line 303 - replace "criterium" with "criterion"

Author’s response: Replaced.

Other Author’s comment: We sincerely thank all the reviewers for their constructive comments that helped to improve the article.
